# Long-Term Changes in Water Body Area Dynamic and Driving Factors in the Middle-Lower Yangtze Plain Based on Multi-Source Remote Sensing Data

Wei Wang [1], Hongfen Teng [1,*], Liu Zhao [2] and Lingyu Han [1]

[1] School of Environmental Ecology and Biological Engineering, Wuhan Institute of Technology, Wuhan 430205, China
[2] School of Geography, Planning and Spatial Sciences, University of Tasmania, Hobart, TAS 7005, Australia
* Correspondence: tenghongfen@zju.edu.cn

**Abstract:** The accurate monitoring of long-term spatial and temporal changes in open-surface water bodies offers important guidance for water resource security and management. In the middle and lower reaches of the Yangtze River, the monitoring of water body changes is especially critical due to the dense population and drastic climate change. Due to the complexity of the physical environment in which the water bodies are located, the advantages and disadvantages of various water body detection rules can vary in large-scale areas. In this paper, we use Landsat 5/7/8 data to extract the area of water bodies in the study area and analyze their spatial and temporal trends from 1984 to 2020 using the Google Earth Engine (GEE) platform. We propose an improved water body extraction rule based on an existing multi-indicator water body algorithm that combines impervious surface data and digital elevation model data. In this study, the performance of the improved algorithm was cross-validated using seven other water body indicator algorithms, and the results showed the following: (1) the rule accurately retained information about the water body while minimizing the interference of shadows on the extracted water body. (2) On the annual scale from 1984 to 2020, the open-surface water body dataset extracted using this improved rule showed that the turning point for the area of each water body type was 2011, with an overall decreasing trend in area before 2011 and an increasing trend in area after 2011, with the exception of special years, such as 1998. (3) The driving mechanism analysis showed that, overall, precipitation was positively correlated with the water body area and temperature was negatively correlated with the water body area. Additionally, human activities can have an impact on surface water dynamics. The key influencing factors are diverse for each water body type; it was found that seasonal water bodies were correlated with precipitation and paddy fields and permanent water bodies were correlated with temperature and urban construction. The accurate monitoring of the spatial and temporal dynamics of open-surface water performed in this study can shed light on the sustainable development of water resources and the environment.

**Keywords:** open-surface water bodies; water indices; shadow; water frequency; the Middle-Lower Yangtze Plain (MLYP)





## 1. Introduction

Water resources are important for aquatic and terrestrial ecosystems, urban development, agricultural production and socio-economics [1–3]. As the main component of water resources, open-surface water bodies mainly include lakes, rivers, wetlands, reservoirs, streams, ponds and partially impounded paddy fields. They account for approximately 3% of the world's land area and are an extremely important component of water resources required for human life and terrestrial ecosystems [4]. The Middle-Lower Yangtze Plain (MLYP) is very rich in water resources, and, in recent years, the implementation of national projects, such as the Three Gorges Project and the South–North Water Diversion, has had

important impacts on the distribution of water resources. Moreover, the climate in the study area is variable, with seasonal precipitation and global warming leading to frequent droughts and floods. Therefore, monitoring the long-term dynamics of water bodies in the MLYP helps to understand the impact of climate change and human activities on the sustainable development of surface water resources in the region.

In recent years, with the development of computer technology and aerospace technology, remote-sensing-based technology has undergone rapid development in the field of surface water body dynamic monitoring and change analysis [5]. Compared to traditional spatial measurement methods, remote sensing technology has the advantages of wide-coverage satellite imagery, low cost and a long data applicability timeframe and can be used to conduct studies in remote areas that are inaccessible to humans [6,7] or even on a global scale [8]. In this study, we used satellite remote sensing to carry out long-term monitoring of a series of water bodies over a large study area.

In widely used satellite datasets for monitoring open-surface water area, common remote sensing data are mainly categorized into three types based on temporal and spatial resolution. Firstly, the Landsat series: since the launch of Landsat 5 in 1984, the temporal and spatial resolutions of Landsat 5/7/8/9 satellite images are 16 days and 30 m, respectively [9]. Secondly, Moderate Resolution Imaging Spectroradiometer (MODIS), which has been operational since 2000, has a temporal resolution of daily and spatial resolutions of 250 m, 500 m, etc. [10,11]. Finally, the Sentinel series: Sentinel 1 has two satellites, Sentinel 1A and Sentinel 1B, launched in 2014 and 2016, respectively, two satellites with a revisit period of 6 days and spatial resolution of 10 m [12]. Sentinel 2 has two satellites, Sentinel 2A and Sentinel 2B, launched in 2015–2017, respectively, two satellites with a revisit period of 5 days and a spatial resolution of 10 m [13,14].

Satellites with long archival times have an advantage when studying long-term surface water dynamics. The Landsat series of satellites has the longest time series (1984-present) and a high spatial and temporal resolution at which most inland water bodies can be accurately captured, and Landsat images have been freely available to the public since 2008 [15]. These above-mentioned advantages render Landsat the best choice for monitoring the long-term dynamics of various water bodies. For example, Pekel et al. used Landsat 5 TM, Landsat 7 ETM+ and Landsat 8 OLI images to achieve long-term mapping of global open-surface water (GSW) bodies from 1984 to 2015 [2]. Xie et al. used Landsat data to describe spatial and temporal patterns of change in major urban lakes in China between 1990 and 2015 [16]. Deng et al. analyzed the long-term changes in the open-surface water bodies in the Yangtze River basin from 1984 to 2018 based on all available Landsat images [17].

The current methods of water extraction involving optical images are divided into two categories according to whether or not they require training samples. Those requiring sample data are machine learning methods, including random forest methods [18], support vector machines [19], deep learning [20,21], etc. In particular, the deep learning method overcomes the problem of unstable results due to complex datasets and thresholds, but it has high computational demands, and this technique is notorious for its "data hunger", which means that it is not attractive for the mapping of a large area of water [14,22].

Rule-based methods do not require known sample data, whereas water-index- and binary-threshold-based methods are widely used due to their simplicity and the accuracy of their results. The most commonly used water body indices in the related studies can be distinguished into two types: single-indicator and multi-indicator thresholding methods. The most common types of single-indicator methods are the normalized difference water index (NDWI) [23], modified normalized difference water index (MNDWI) [24] and automatic water extraction index (AWEI), which uses two modes: shaded images with dark surfaces (AWEIsh) and shadowless images (AWEInsh) [25]. With the expansion of cities, building shadows have a great impact on water extraction [26,27]. In this regard, some scholars proposed the robust multi-band water index (MBWI) [28], and Wu et al. proposed the two-step urban water index (TSUWI), in which USI can eliminate the interference of

urban shadows with high-resolution satellites [29]. However, the use of a single water body index may lead to confusion in distinguishing water bodies and vegetation. In recent years, the enhanced vegetation index (EVI) [30] < 0.1 and MNDWI > normalized difference vegetation index (NDVI) [31] or MNDWI > EVI, combining the vegetation index and water index, have been widely used to extract water bodies [4,32–36]. Accurate results were obtained in these studies, but some studies found that the method extracted the water bodies with large errors. For example, in the upper reaches of the Yellow River, the method extracted incomplete water bodies, and, in this paper, water bodies were extracted by the MNDWI > NDVI or MNDWI > EVI method after removing the EVI > −0.1 condition [9]. However, the extraction effect of the MNDWI > NDVI or MNDWI > EVI method has not been validated in other areas. The method is also not applicable in the Yangtze River basin. Deng et al. constructed a multi-index water detection rule (MIWDR) using MNDWI, AWEI, NDVI and EVI, and the method achieved good results for the Yangtze River basin [17], Guangdong–Hong Kong–Macao Greater Bay [37] and Dongting Lake [38]. However, the method could not fully address the influence of urban buildings, and it also led to the misclassification of narrow urban rivers.

In studies of extracting surface water bodies, using only the index method often cannot achieve good extraction results. In this case, using some products or data can remove interfering factors. For example, terrain shadows are easily misclassified as water bodies. Yang et al. [14]. used digital elevation models (DEMs) to eliminate the influence of terrain shadows, and two hydrological terrain models, HAND [39] and HydroSHEDS, were utilized to eliminate mountain shadow effects. In addition, buildings and building shadows are also easily mistaken for water bodies. Zhou et al. [9] used the Global Human Settlement Layer (GHSL) and building grid data to remove building pixels that are easily mislabeled as water pixels. Similarly, impervious surface data can achieve the same effect of removing building pixels, such as the global artificial impervious area (GAIA) [40], with a higher temporal resolution.

In this study, water bodies were extracted using the water index method based on Landsat satellite images in a long time series over a wide-ranging area, and it was difficult for the traditional research platform to handle many satellite images. In recent years, many remote sensing big data platforms have emerged, such as Google Earth Engine (GEE), PIE Engine, Microsoft Planet Compute, etc. Among them, GEE is the most widely used. It is a cloud-based geospatial processing platform that stores the petabytes of free satellite imagery. The GEE platform also provides Python and JavaScript application programming interfaces (APIs) so that researchers can write code to process their data quickly [41,42]. GEE is more convenient and time-efficient than traditional remote sensing software. In this study, Landsat data for water extraction, Sentinel 2 data for accuracy verification, GAIA data and DEM data for shading removal were all gathered directly from the platform.

The difficulty and effectiveness of the water extraction methods were considered comprehensively, and the combination rule of multiple indices was chosen to extract the water bodies. The objectives of this study were as follows: (i) to develop an improved method and test its performance compared to existing methods, and (ii) use this new method to carry out a trend analysis of water body fluctuations in the study area. After adding impermeable surfaces and DEM to remove shadows, the improved method is significantly better than the existing method in mapping water bodies in the MLYP.

## 2. Materials and Methods

### 2.1. Study Area

The MLYP is located in the central and eastern parts of China (24°29′N–35°20′N, 108°22′E–123°10′E), east of the Wushan Mountains, west of the Yellow Sea and East China Sea and south of the Qinling and Huai River (Figure 1). The administrative division includes seven provinces and cities, namely Hubei, Hunan, Anhui, Jiangxi, Jiangsu, Zhejiang and Shanghai, with a total area of 925,000 km$^2$. There are various types of landforms in the area, including plains, mountains and hills and basins between mountains. The water resources

in the MLYP are the most abundant in China. The Yangtze River has the largest average runoff among the various rivers, being the third largest river in the world. Other rivers with relatively large annual runoffs include the Xiangjiang, Ganjiang and Huaihe. Four of the five largest freshwater lakes in China are situated within the region, including Poyang Lake, Dongting Lake, Taihu Lake and Hongze Lake. In recent years, with population growth and rapid economic development, the surface water has undergone drastic changes, and droughts and floods have occurred frequently, causing casualties and economic losses.

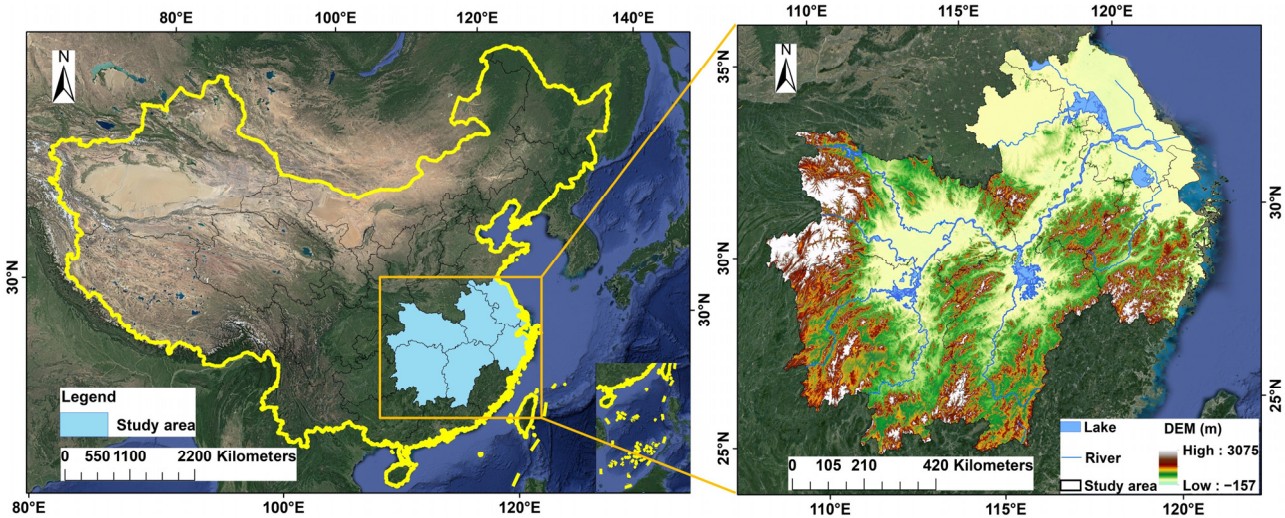

**Figure 1.** The location of the study area.

### 2.2. Datasets

### 2.2.1. Long-Time-Series Landsat Data

In this study, the surface reflection (SR) products of the United States Geological Survey Landsat images (Landsat 5/7/8) were utilized to extract the surface water area in the MLYP. Landsat 5 and 7 surface reflectance datasets were generated using the Landsat Ecosystem Disturbance Adaptive Processing System (LEDAPS) algorithm, and Landsat 8 surface reflectance datasets were generated using the Landsat Surface Reflectance Code (LASRC) algorithm [43]. Then, these datasets were algorithmically and atmospherically corrected, radiometrically calibrated and FLAASH atmospherically corrected by the USGS [44] and uploaded to the GEE platform. We selected the blue, green, red, NIR, SIR1 and SIR2 bands of the Landsat 5/7/8 surface reflectance products for the index threshold method of water extraction. The time distribution of the Landsat images and the number of images acquired each year are shown in Figure 2. All the Landsat (TM, ETM+ and OLI) surface reflectance images were obtained from 16 March 1984 to 31 December 2020. The Landsat images from 1984 to 1986 were not sufficient to cover the study area; hence, we chose to combine the images from 1984 to 1987 as the first period of data during the study period.

Figure 2a shows the total number of observations per pixel from 1984 to 2020. The acquisition of high-quality satellite imagery is critical for the generation of an annual water map. In this paper, we use the "CLOUD_COVER" attribute and "pixel_qa" band in Landsat 5/7/8 SR [45] to mask satellite images that are not functional due to invalid pixels, clouds, cloud shadows, snow and ice, etc. The spatial distribution of the annual number of good observations across the study area from 1984 to 2020 can be found in Figure 2c.

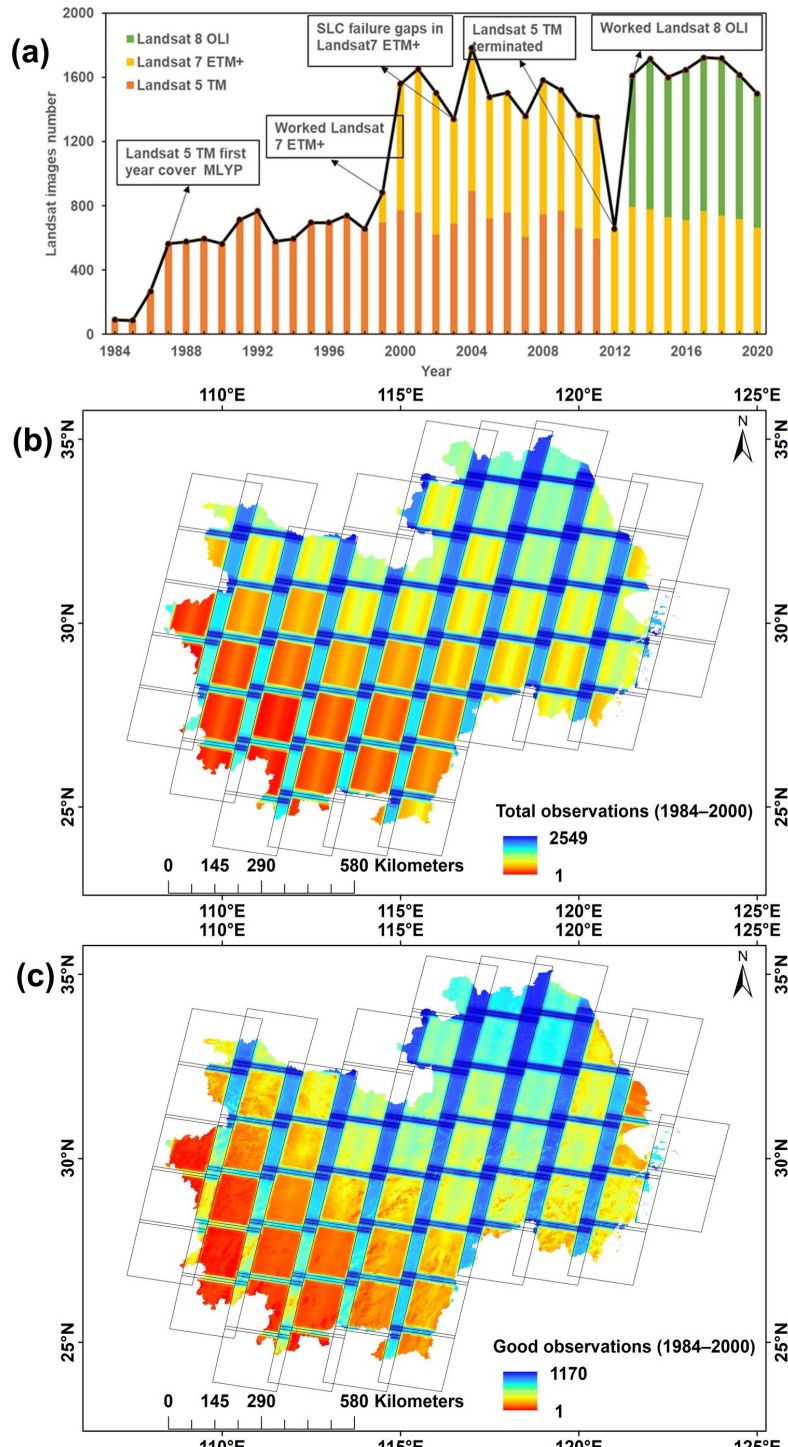

**Figure 2.** Landsat images used in this study: (**a**) number of scenes per year from different Landsat sensors; (**b**) spatial distribution of the total number of Landsat images; (**c**) spatial distribution of the number of good Landsat images.

### 2.2.2. Auxiliary Data

1. Sentinel 2 Multispectral Instruments (MSI) Images

Sentinel 2 images with a revisit period of 10 days and a spatial resolution of 10 m were acquired in 2020 to generate water and non-water samples in order to verify the accuracy of the methods used to extract the water in this study.

2. The global artificial impervious area (GAIA)

The global artificial impervious area (GAIA) offers annual change information on the global impervious surface area from 1985 to 2018 at a 30 m resolution. The change from pervious to impervious was determined using a combined approach of supervised classification and temporal consistency verification. Impervious pixels are defined as those that are impervious above the level of 50%. The year of the transition (from pervious to impervious) can be identified using the pixel value, ranging from 34 (year 1985) to 1 (year 2018) [40]. These data are used as auxiliary data for the removal of urban noise from the water bodies extracted from 1984 to 2020. The time interval of the study reported in this paper was 1984–2020, and the Landsat images from 1984 to 1987 were taken as the first period, as mentioned above, and the impermeable surface of 1987 was chosen as the starting year. The products were not available in 2019 or 2020 and the data from 2018 were used for these two years. Compared to the Global Human Settlement Layers, for the Built-Up Grid (GHSL) product, which is 10 or even 15 years apart, a gap in the data of one or two years is not significant.

3. Global Surface Water (GSW) data

The Joint Research Center (JRC) [2] global surface water maps were generated using approximately 4.5 million Landsat images with a spatial resolution of 30 m, which have been available since 1984. At the JRC, each pixel was individually classified into water/non-water using an expert system. The JRC global surface water layer v1.3 for 1984–2020 and the JRC annual water classification history v1.3 were selected as validation data for comparison with the open-surface water extracted in this study.

4. HAND and GMTED2010

Notably, terrain shadows are easily misclassified as water bodies [14]. Two terrain models, the Height Above the Nearest Drainage (HAND) [39] and Global Multi-Resolution Terrain Elevation Data 2010 (GMTED2010), were used to mask the effects of terrain shadows. In this paper, GMTED2010 is used in combination with the azimuth and zenith angle to mask the shadows of hills, and HAND is used to mask terrain shadows on flat areas.

5. Precipitation and temperature data

The two datasets on precipitation and temperature were obtained from the 'Chinese Meteorological Elements Day-by-Day Station Observation Dataset' of the Resource and Environment Science and Data Center of the Chinese Academy of Sciences (https://www.resdc.cn, accessed on 1 December 2022), a dataset obtained from daily value observations of more than 2400 meteorological stations in China. There are 507 meteorological stations in our study area, and the daily mean temperature and daily 20–20 h cumulative precipitation from these meteorological stations were selected for the period 1984–2020. In this paper, we attempt to analyze the water body area changes on an annual scale, and the climate data should also be treated as an annual scale.

We summarize the daily average temperature for the whole year and then divide it by one year to obtain the annual average temperature and summarize the daily 20–20 h cumulative precipitation for the whole year to obtain the annual total precipitation.

*2.3. Methodology and Flowchart*

In order to clearly express the workflow, we created a general flowchart framework consisting of three steps (Figure 3). Firstly, we collected remote sensing data on the GEE platform and pre-processed the images. Secondly, we performed a comparison and validation of each water detection rule, followed by the selection of the best rule and water body mapping. Finally, we performed accuracy verification, an analysis of the spatial and temporal dynamics of water frequency and drive mechanism analysis. The details are described in the following sections.

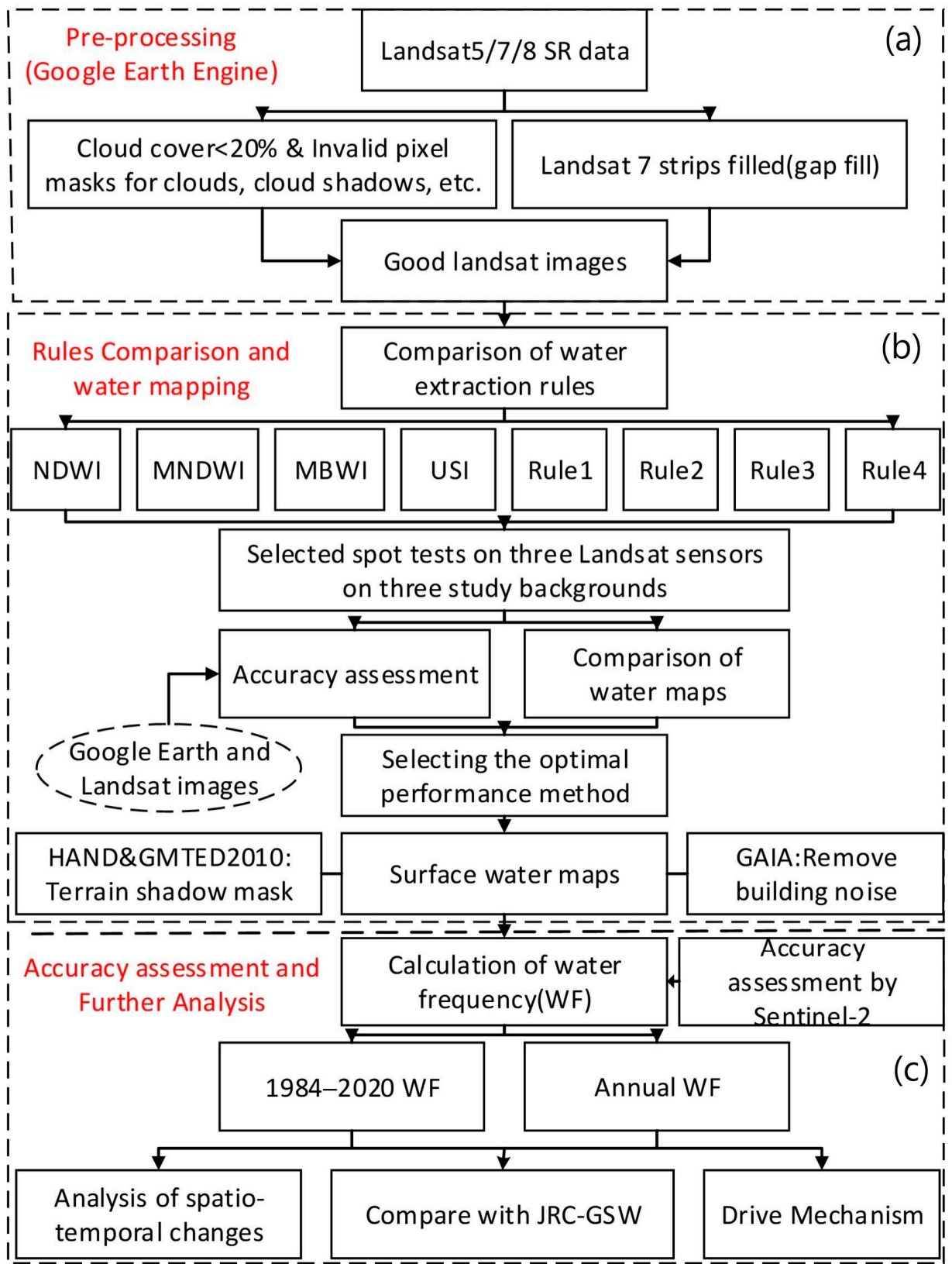

**Figure 3.** The framework of this study.

2.3.1. Data Pre-Processing

As mentioned above, Landsat images have removed cloud, cloud shadow and other undesirable pixel interference. Due to the failure of Landsat7 after 2003, the generated

images have obvious stripes. Thus, here, we used the GapFill [46] function on the GEE platform to interpolate the stripes so as to prevent them from affecting the water body extraction. To generate a cloud-free synthetic image of the study area, we mean-synthesized the images of the study area on a year-by-year basis. In this paper, we sought to verify the accuracy of the results with Sentinel 2. The same method as that employed for Landsat was used to mean-synthesize the Sentinel 2 images after removing clouds and other disturbances.

2.3.2. Methods of Water Extraction Based on the Water Detection Rule

In our study, water bodies were extracted based on the water index and threshold method, a method that is commonly used to extract water bodies from remote sensing images, with the advantages of simplicity, efficiency, automation and accuracy. As outlined in the Introduction, the existing commonly used water index methods have some deficiencies. Zhou et al. used MNDWI > NDVI or MNDWI > EVI rule to extract water bodies in the upper Yellow River with good results, and we tested the method in our study area. In order to find out the spectral differences between water bodies and other background pixels, five categories of pixels were analyzed: water bodies and mountain shadows, vegetation, bright buildings, dark buildings and shadows. For each category, 600 pixels were selected, and a total of 3000 pixels were selected from the Landsat images in Table S1. The box plots of NDWI, MNDWI-NDVI and MNDWI-EVI values were drawn using these five categories of pixels (Figure 4).

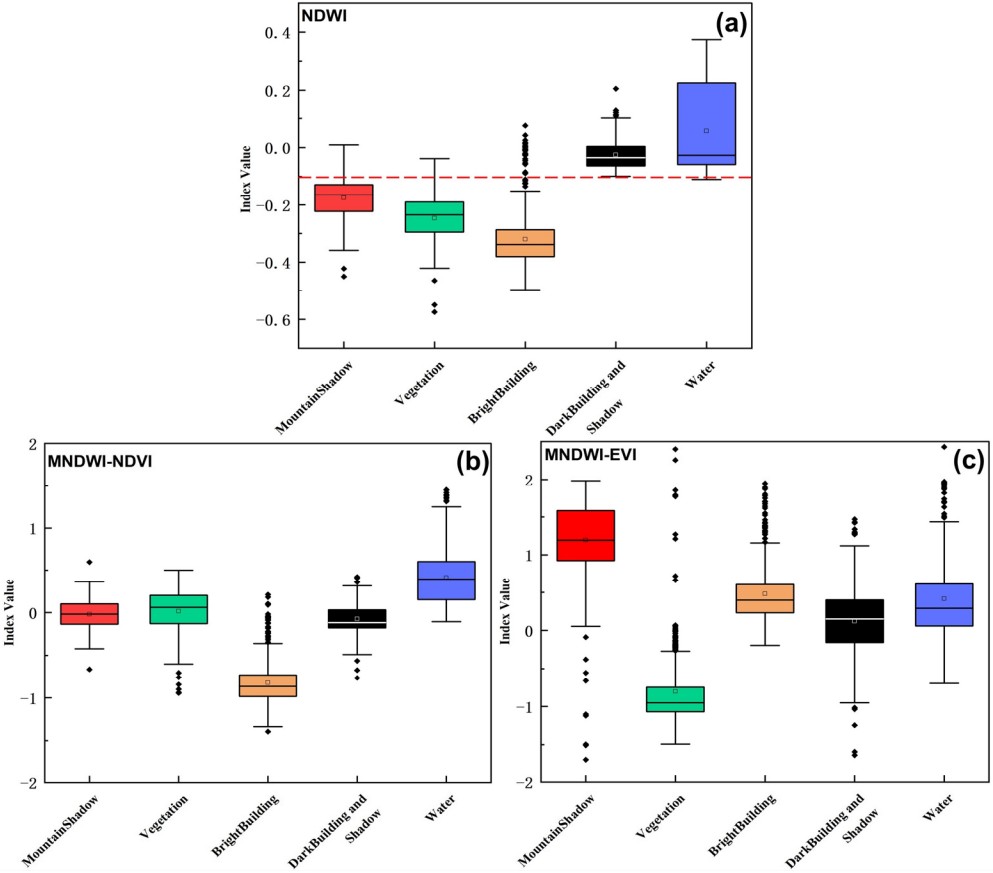

**Figure 4.** The box plot of five types of pixel index values for mountain shadows, vegetation, bright buildings, dark buildings and shadows, and water. (**a**) NDWI. (**b**) MNDWI-NDVI. (**c**) MNDWI-EVI.

As shown in Figure 3b,c, the MNDWI > NDVI or MNDWI > EVI rule caused significant interference in the extraction of water bodies. Mountain shadow pixels could not be completely eliminated, vegetation could not be entirely removed and building shadows and various types of buildings also had some impact on water body extraction. The

commonly used method of NDWI > 0 was criticized for its poor performance in extracting small water bodies. We chose some small water body pixels, as shown in Figure 3a, with all water pixels having NDWI values greater than −0.1. When the NDWI threshold was set to >−0.1, mountain shadows, vegetation and highlighted buildings could be effectively eliminated. However, NDWI > −0.1 was not as effective as MNDWI > NDVI or MNDWI > EVI in rejecting dark building and shadow pixels. In summary, we proposed a new and improved combinatorial algorithm where a pixel was classified as a water body when it satisfied the following conditions: NDWI > −0.1 and MNDWI > NDVI or MNDWI > EVI. The indices involved in this algorithm are defined as follows.

$$\text{NDWI} = \frac{(\rho_{GREEN} - \rho_{NIR})}{(\rho_{GREEN} + \rho_{NIR})}, \tag{1}$$

$$\text{MNDWI} = \frac{(\rho_{GREEN} - \rho_{SWIR1})}{(\rho_{GREEN} + \rho_{SWIR1})}, \tag{2}$$

$$\text{NDVI} = \frac{(\rho_{NIR} - \rho_{RED})}{(\rho_{NIR} + \rho_{RED})}, \tag{3}$$

$$\text{EVI} = 2.5 \times \frac{(\rho_{NIR} - \rho_{RED})}{(\rho_{NIR} + 6 \times \rho_{RED} - 7.5 \times \rho_{BLUE} + 1)}, \tag{4}$$

where $\rho_{GREEN}$, $\rho_{NIR}$, $\rho_{SWIR1}$, $\rho_{RED}$, $\rho_{BLUE}$, represent the values of the green band, near infrared band, shortwave infrared band1, red band, blue band in the Landsat Images.

In order to extract water bodies with the highest accuracy in the MLYP, we compared the advantages and disadvantages of eight water body extraction methods, including seven commonly used methods at present and the new method in this paper (Table 1), and the results of the comparison were used to determine the best algorithm for extracting water bodies in the MLYP. To maximize the noise removal, we used the HAND and SRTM auxiliary datasets to mask the topographic shadows and the GAIA impervious surface dataset to remove urban building pixels that could easily be misclassified as water body pixels. The surface water map we obtained with the best algorithm and the impervious surface data was intersected and inverted. It was then used to remove the urban noise and finally generate water body products of the study area for the 1984–2020 time series.

**Table 1.** Water mapping rules.

| Type | Number | Water Indices | Bands | Standard | Literature |
|---|---|---|---|---|---|
| Single Index | 1 | NDWI | G, NIR | NDWI > 0 | [23] |
| | 2 | MNDWI | G, SWIR1 | MNDWI > 0 | [24] |
| | 3 | MBWI | B, G, R, NIR, SWIR1, SWIR2 | MBWI > 0 | [28] |
| | 4 | USI | B, G, R, NIR | USI > 0.1 | [29] |
| Multiple Indices | 5 (rule1) | E-MVI | B, G, R, NIR, SWIR1 | EVI < 0.1 and (MNDWI > NDVI or MNDWI > EVI) | [4] |
| | 6 (rule2) | MVI | B, G, R, NIR, SWIR1 | (MNDWI > NDVI or MNDWI > EVI) | [9] |
| | 7 (rule3) | A-MVI | B, G, R, NIR, SWIR1, SWIR2 | (AWEInsh − AWEIsh > 0.1) and (MNDWI > NDVI or MNDWI > EVI) | [17] |
| | 8 (rule4) | N-MVI | B, G, R, NIR, SWIR1 | (NDWI > −0.1) and (MNDWI > NDVI or MNDWI > EVI) | This study |

Note: B, G, R, NIR, SWIR1, and SWIR2 correspond to the bands of blue, green, red, near-infrared, shortwave infrared 1, and shortwave infrared 2, respectively.

### 2.3.3. Accuracy Assessment

The accuracy validation reported in this paper is divided into two steps. The first step selected Sentinel 2 to validate the accuracy of the method in this paper. Compared to the 30 m spatial resolution of Landsat 5/7/8, the 10 m spatial resolution of Sentinel 2 provides a clearer and more accurate extraction of the mixed pixels of water–land boundaries and some water bodies, such as creeks and ponds. For this paper, we selected the 2020 Sentinel 2 extractions to validate the 2020 Landsat extractions of water bodies The N-MVI method was used to extract water bodies based on the 2020 Sentinel 2 images with the building noise and mountain shadows removed, and water and non-water samples were selected by stratified sampling of the generated water map. To ensure the maximum reliability of the assessment results, some water samples were visually selected from small rivers and water–land boundaries and some non-water samples were selected from areas near the banks of large rivers. A total of 5000 water samples and 5000 non-water samples were generated, and these sample points were added to the 2020 Landsat-generated water body map. Confusion matrices for the UA, PA, OA, Matthews correlation coefficient (MCC), kappa coefficient and F1 Score were calculated from these samples.

In the second step, to compare the accuracy of the N-MVI method in this paper with other existing methods, we randomly select several sample areas, which included typical features of the study area, such as urban areas, mountainous areas, rice fields, etc., and then select images with different sensors using Landsat 5/7/8. A total of 9 images (Table S1) were selected in this way so as to compare the accuracy of water extraction using the 8 methods with different Landsat sensors in different study contexts. Sample points were randomly generated from 9 Landsat tiles with less than 1% cloudiness in the study area. Water and non-water samples were identified through the visual interpretation of Landsat images and Google Earth images, and some typical sample points were added manually. Finally, a total of 1000 water samples and 3000 non-water samples were generated from the 9 Landsat tiles and the water bodies extracted using the 8 methods were compared with the visually interpreted samples. The user's accuracy (UA), producer's accuracy (PA), overall accuracy (OA) and kappa coefficient of the different methods are shown in Table S2.

### 2.3.4. Calculation and Change Analysis of Open-Surface WF

Before analyzing the changes in dynamic classification of the open-surface water bodies, time series water body frequencies were calculated for the study area from 1984 to 2020 and for each year, respectively. The WF was calculated as:

$$\text{WF} = \frac{\sum\limits_{i=1}^{N} (\varepsilon_i = 1)}{N} \times 100\%, \tag{5}$$

where $\varepsilon$ represents the corresponding cell value of the $i$th open-surface water body map (1 for water, 0 for non-water), accumulated in GEE with 'sum', where N denotes the total number of valid observations of Landsat image elements in a given period, calculated as 'count' in GEE.

The range of WF is from 0% to 100%, and any pretzel noise pixels are masked by pixels with WF > 0. In a previous study [4], 5% and 75% were used as breakpoints for the extraction of ephemeral, seasonal and permanent water bodies in the US. In this paper, 5%, 25%, 75% and 95% are used as breakpoints (Table 2). Specifically, pixels with 95% < WF $\leq$ 100% were classified as permanent water bodies; pixels with 75% < WF $\leq$ 100% were classified as year-long water bodies; pixels with 25% < WF $\leq$ 75% were classified as seasonal water bodies; water pixels with 5% < WF $\leq$ 25% were classified as temporary water bodies, which are inclined to become seasonal water bodies; and water pixels with 0 < WF $\leq$ 5% were classified as temporary water bodies.

**Table 2.** Classification rules for water types based on the water frequency.

| ID | Classification Rules | Description | Name |
|----|---------------------|-------------|------|
| 1 | $0 < WF \leq 5\%$ | Temporary water bodies | TWB |
| 2 | $5\% < WF \leq 25\%$ | Temporary water bodies that are inclined to seasonal water bodies | TWBS |
| 3 | $25\% < WF \leq 75\%$ | Seasonal water bodies | SWB |
| 4 | $75\% < WF \leq 100\%$ | Year-long water bodies | YLWB |
| 5 | $95\% < WF \leq 100\%$ | Permanent water bodies | PWB |

## 3. Results

### 3.1. The New and Improved Water Body Extraction Method in This Paper

3.1.1. The Application of the Water Body Extraction Method in MLYP

We applied the method described in this article to generate a water map of MLYP in 2020, as shown in Figure 5. Figure 5a showed the water map generated using the newly improved multi-index algorithm. It could be seen from the locally enlarged map that the algorithm could not completely remove the water extraction errors caused by some mountain shadows and building shadows. Figure 5b showed the water map of Figure 5a masked with DEM data, which effectively removed water extraction errors caused by mountain shadows. The water map of Figure 5a and the impervious surface data (GAIA) were intersected and inverted, and then we obtained Figure 5c, where all building shadows in the Wuhan urban area have been removed, as observed from the locally enlarged map. By following these three steps, we finally obtained the water map of MLYP in 2020.

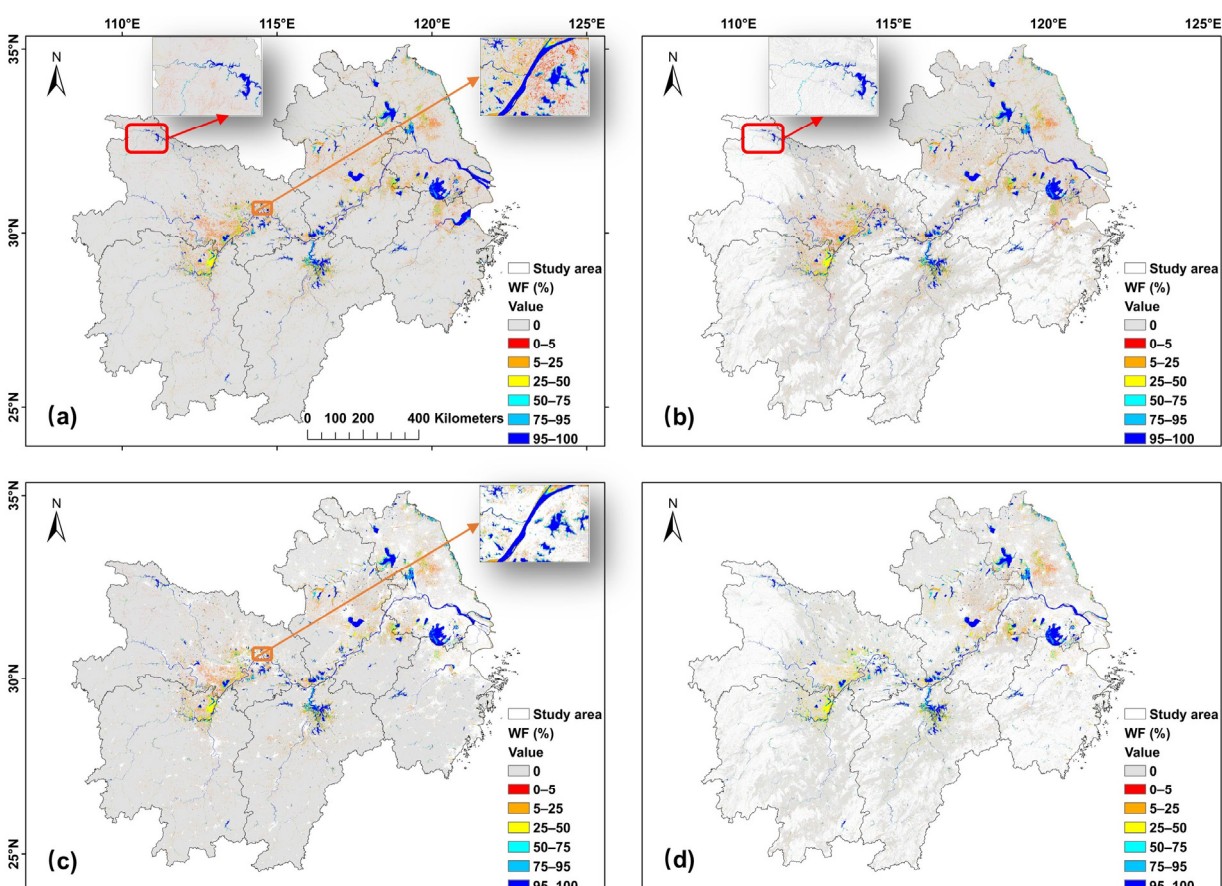

**Figure 5.** The flow map of the water extraction method in MLYP in 2020. (**a**) The water extraction map of N-MVI index method; (**b**) the water map of Figure 5a masked with DEM data; (**c**) the water map of Figure 5a and the impervious surface data were intersected and inverted; (**d**) the final generated water map.

### 3.1.2. Accuracy Assessment of the Extracted Open-Surface Water in the MLYP Using Sentinel 2 Images in 2020

As described in Section 2.3.3 Accuracy assessment, we validated the water map of Landsat 2020 generated by the method of this paper using Sentinel 2 images of 2020. The same method was used regardless of which option was chosen to differentiate between the water body types in this study. The largest water body was chosen as an example, and the results are shown in Table 3. According to the confusion matrix, the producer accuracy of the water extraction results was 96.26%, the user accuracy was 98.91%, the kappa coefficient was 95.18%, the overall accuracy was 97.6%, the MCC was 94.77% and the F1 was 97.57%. These data indicate that the results for the water bodies extracted from the MLYP area were obtained with particularly high accuracy and can be applied for further analysis.

**Table 3.** Accuracy assessment of the sample points.

| | Samples | Sentinel | | Total | User Accuracy | Overall Accuracy | Kappa Coefficient |
|---|---|---|---|---|---|---|---|
| | | Water | Non-Water | | | | |
| Landsat | Water | 4813 | 53 | 4866 | 98.91% | | |
| | Non-water | 187 | 4947 | 5184 | 95.43% | | |
| Total | | 5000 | 5000 | 10,000 | | 97.6% | 95.18% |
| Producer's accuracy | | 96.26% | 98.94% | | MCC = 94.77% F1 = 97.57% | | |

Since our extraction method used auxiliary data, it was not capable of proving that the new and improved multi-index algorithm proposed in this paper is the best algorithm for extracting water bodies in the MLYP. Therefore, we selected seven existing commonly used index methods to compare with the N-MVI index method to further verify the applicability of the new and improved multi-index algorithm in the MLYP.

### 3.2. Comparative Analysis of Different Water Extraction Methods

### 3.2.1. Results of Different Water Index Methods in Built-Up Areas

The pixels of urban objects are very complex, and there are many pixels that are easily confused with water, such as building shadows, dark buildings and highlighted building surfaces, resulting in overestimated results during the extraction of large spatial water bodies. Due to the change in solar altitude angle, the shadows of urban buildings also change. Thus, images from different months were selected for the cities and mountains with shadows in order to analyze the effect of shadows on the water bodies extracted by each method, and the information for the selected images is shown in Table S1. The water maps of eight indicators at three time points are shown in Figure 6a, with the addition of water maps for the intersection of the inverse impervious surfaces after applying the N-MVI method to urban areas. When using MBWI for urban areas and setting the conventional threshold value as greater than 0, the water body was subject to very large error and the threshold value of MBWI was set to −1000 by debugging.

From Figure 4a, it can be seen on the time 3 image with a high solar height that, in contrast to E-MVI and MVI, the other methods suppress the urban shadows well, especially MBWI. According to Table S2, the UA is 97.2%, but the PA is only 70%, and removing the shadows sacrifices the ability to detect water. E-MVI misses the largest number of water bodies. Its PA = 60%, and the large rivers and lakes have different degrees of omission, a result that is the same as the findings of previous studies. The largest PA values are those for MNDWI, MVI and N-MVI, with almost no omission, and the PA of MVI is the lowest of the three. Although there was less shade in August, the vegetation index could not remove the highlighted buildings, but N-MVI could remove the highlighted buildings well following the addition of an NDWI greater than −0.1 to its conditions. The OA and kappa of MNDWI are known to be better than those of N-MVI extraction, but, after intersecting the inverse impermeable surface on the N-MVI water map, its OA and kappa were greater than those of MNDWI. At time 2, the maximum PA of MVI and N-MVI is 98.31%. The

maximum UA of USI is 100%, but its PA is only 25.42%, which indicates that USI is unstable and the water extraction effect will be significantly improved when the threshold is set to greater than −0.1 on this tile, which, in turn, indicates that USI does not have a fixed threshold. Therefore, one needs to set different thresholds for different sensors at different times, a practice that cannot be used in the long-term study of water bodies. For the tile of time 1, MNDWI has the highest PA, but UA has lower-value results compared with the other two times, and the suppression of urban shading decreases in a step-by-step manner for these three times, indicating that the shading noise of MNDWI is more significant in the months with a lower solar height. Previously, A-MVI was used to extract water bodies in the Yangtze River basin, but, from Figure 4a and Table S2, one can observe that its effect is poor and the water bodies are left unextracted in time 2 and time 3, while the water bodies are well extracted in time 1, with a PA of 71.1%. Moreover, we can observe the failed extraction of the fine rivers, but the UA is very low, and, in addition to the building noise, there are also effects of urban hill shadows and vegetation, the misplacement noise of which the method cannot remove. However, the N-MVI method can solve these noises very effectively. This problem will be described in detail for the mountainous areas and rice fields. We obtained the average OA, kappa, UA and PA of all the rules, and we determined that the best method for retaining the information on the water bodies is the MNDWI method, with an average PA = 95.83%. The best method for removing all the noise is the USI method, with an average UA = 92.3%. Although N-MVI does not remove building shadows, it has the highest average OA and average kappa of all the methods, with 90.22% and 77.67%, respectively. After adding impermeable surfaces to the N-MVI water map for the purpose of processing, the OA and kappa reached the maximum value at each time point.

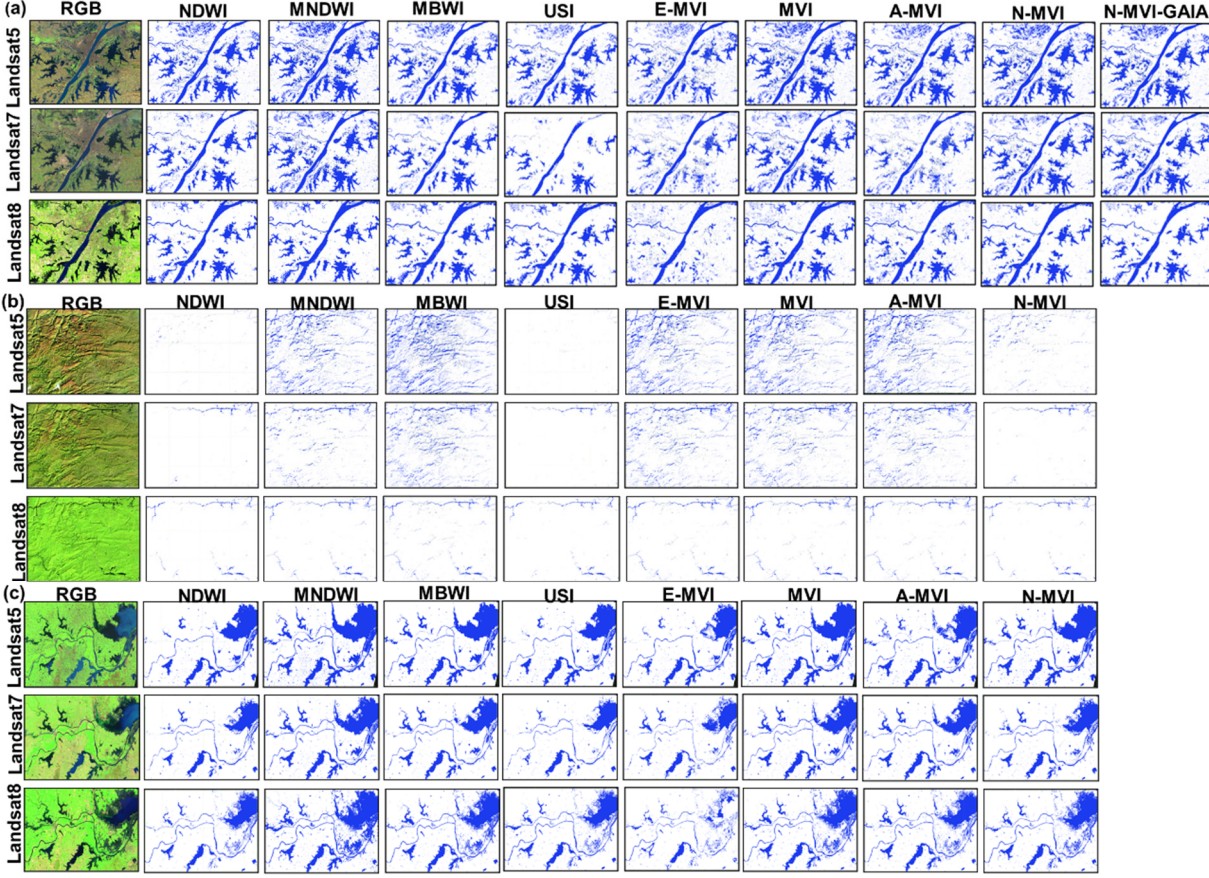

**Figure 6.** Performance of the 8 water body algorithms applied to different images of Landsat 5/7/8 in the city (**a**), mountainous areas (**b**) and rice fields (**c**). The RGB images of Landsat correspond to the shortwave infrared-1, near infrared and red bands, respectively.

### 3.2.2. Results of the Water Index Methods in Areas with Shadows of Mountains

The shading of mountains in the study area is an issue that cannot be ignored, and the effect of mountain shading on water bodies is often discussed in large-scale water extraction studies. Thus, in this section, we explore the results of the different water index methods for water extraction in mountainous areas.

It was mentioned above that shading has a great influence on water extraction. Thus, Landsat tiles from different months were chosen for the mountainous areas so as to compare the results of the eight rules. As shown in Figure 6b and Table S1, at time 4,5, it is obvious that, with five methods, namely MNDWI, MBWI, E-MVI, MVI and A-MVI, the degree of mountain shading is serious, whereas, at time 6, in contrast to the MNDWI method, the other four methods still result in mountain shading that cannot be removed. At time 4, N-MVI cannot completely remove the mountain shadow, and USI is the most significant for each time period of water leakage lifting. As shown in Table S2, illustrating the mean value of the accuracy assessment values for the three time periods, MNDWI and MVI retained the best water body information, with PA = 97%. NDWI removed the noise of mountain shadows the most effectively, with UA = 96.83%. The OA = 94.58% and kappa = 86.86% of N-MVI are the largest of the eight methods. NDWI extracts mountain water effectively, and the difference between NDWI and N-MVI is mainly related to the fine water bodies, which NDWI cannot identify accurately.

### 3.2.3. Results of the Water Index Methods in Fields

The rice field was selected as the main validation area to test the effect of vegetation on the extracted water bodies. Thus, April was chosen for times 7, 8 and 9 (Table S1), marking the month in which the rice was in the growth period and could be clearly observed. It can be seen in Figure 6c that E-MVI shows a highly obvious absence. Comparing E-MVI with the MVI method, we found that the problem is that EVI < 0.1. This condition is set to remove vegetation pixels that are easily confused with water, but many water pixels that meet this condition are also removed in the MLYP, causing the extracted water results to have large errors with respect to the actual value. The A-MVI method misses the water in time 7 and the PA is only 75.56%. Moreover, the method does not remove vegetation in times 8 and 9. NDWI, USI, E-MVI and N-MVI remove the vegetation pixels effectively and the average UA value is greater than 90%. The average PA value of MNDWI = 98.78% was the largest of all methods; however, its average UA value was the smallest. USI and NDWI have different degrees of water body leakage when detecting small, discrete water bodies, such as ponds and paddy fields. From a comprehensive viewpoint, N-MVI removes the vegetation noise and enhances the water body details with the best effects. Its OA = 96.71% and its kappa = 91.43%. When verifying the accuracy of these nine Landsat tiles, the average OA, kappa of N-MVI was the highest in each study context, but the method still mis-extracted building shadows and some mountain shadows. Thus, it is necessary to combine impervious surface data and mountain shadow data from the digital elevation model on the basis of this method in order to obtain the best water extraction results.

### *3.3. Dynamics of the Open-Surface Water in the MLYP*

### 3.3.1. Spatial Changes in the Open-Surface Water in the MLYP from 1984 to 2020

The spatial distribution of the WF in the MLYP from 1984 to 2020 is shown in Figure 7. The open-surface water bodies consist mainly of lakes, rivers, reservoirs, wetlands, ponds, mudflats and paddy fields. From the table of latitude and longitude for the area in Figure 7f,g, we can see that the water bodies in the study area are mainly distributed on a longitude of 111–121 °C and latitude of 28.3–33.2 °C, and there are four peaks in longitude at Dongting Lake, Poyang Lake, Hongze Lake and Taihu Lake, as well as the peaks in latitude on both sides of Yangtze River.

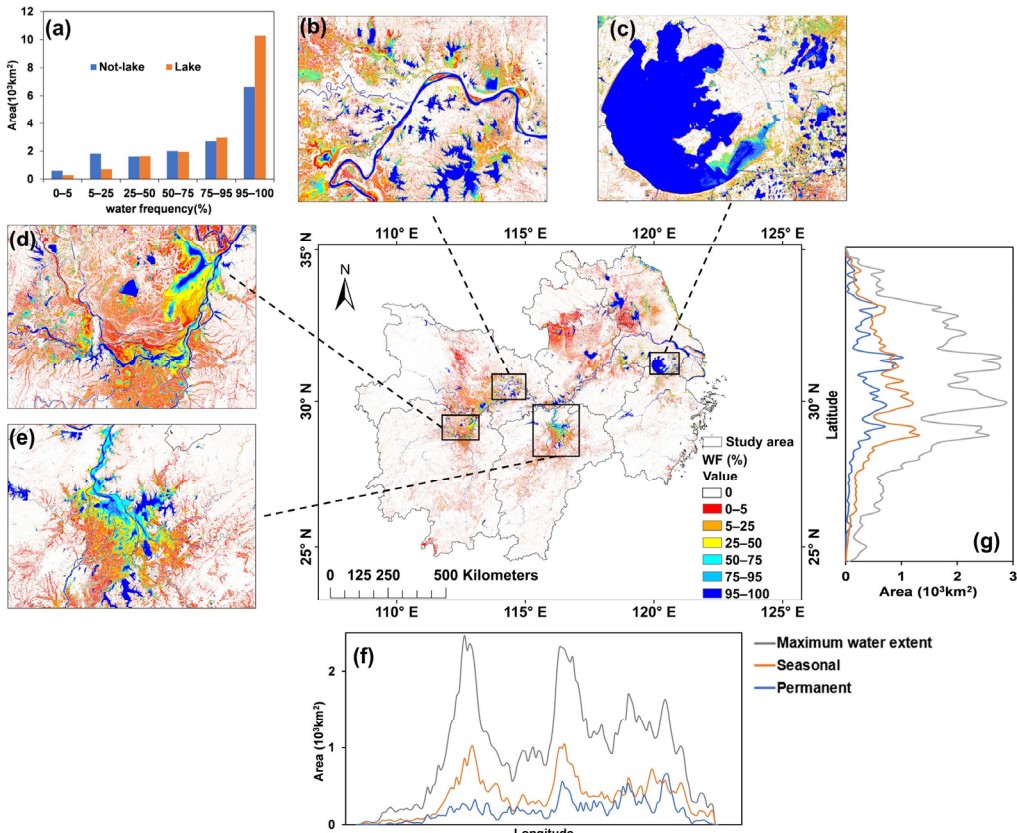

**Figure 7.** Maps of the overall and locally amplified spatial distributions of open-surface water frequencies in the MLYP from 1984 to 2020. (**a**) Comparison of lake and non-lake water frequencies. (**b**) Wuhan Lakes Group, (**c**) Taihu Lake, (**d**) Dongting Lake and (**e**) Poyang Lake (**f**) with 0.1° longitude summaries of the open-surface water area and (**g**) with 0.1° latitude summaries of the open-surface water area.

From the different proportional frequency compositions of the open-surface waters, we can see that, when WF > 0, the largest open-surface water body in the study area is 138,200 km², accounting for 15% of the whole MLYP. By importing the vectors of large lakes, large reservoirs and first-class rivers, it can be observed that the frequency of large water bodies is basically above 0.95, accounting for 99% of the frequency of 0.95 water bodies in the study area. In Figure 7a, we can see that the lakes are larger than the non-lakes in almost all the water body frequencies and smaller than the non-lakes in only <25% of the water body frequencies, which shows that lakes are pivotal in the study area.

The frequency area of each water body, in order of size, is 0 ≤ WF < 5%, 5% < WF ≤ 25%, 75% < WF ≤ 100%, 25% < WF ≤ 75%, 95% < WF ≤ 100%. Among them, the area of temporary water bodies with WF < 5% is 61,070 km², accounting for 44.2% of the total water bodies. It is mainly composed of temporary water bodies, such as rainwater, flooding areas, etc. The area of water bodies with 5% < WF ≤ 25% is 28,078 km², accounting for 20.3% of the total water bodies, mainly consisting of paddy fields, wetlands, swamps, etc. The area of permanent water bodies with 95% < WF ≤ 100% is 17,096 km², accounting for 12.4% of the total water bodies, consisting of large lakes, reservoirs and the main streams and first and second tributaries of the Yangtze River, the latter being one of the important components of open-surface water bodies. This area existed year-round during the study period. Meanwhile, the year-long water body area with 75% < WF ≤ 100% accounts for 19.3% of the total water bodies, existing continuously throughout the year, while the seasonal water with 25% < WF ≤ 75% accounts for 16.2% of the total water bodies, mainly consisting of large water body boundaries, paddy fields, small rivers and ponds, coastal fishing grounds, etc. It existed for between one and three seasons throughout the year,

located in a transitional position between the year-long water body and the temporary water body.

### 3.3.2. Temporal Changes in Open-Surface Water in the MLYP from 1984 to 2020

The trend in the area changes for each water body type and each water body frequency can be seen in Figure 6a,b. For the three water body types, PWB, YLWB and SWB+YLWB, in Figure 6a, the water area changes are roughly divided into three stages. The change trend is not obvious before 1998, and the overall trend is a decreasing one. In 1998, when a major flood occurred in the Yangtze River basin, the areas of all types of water bodies increased significantly. After 1998, the areas of these three types of water bodies generally showed a decreasing trend, and, in 2011, the water area decreased to the lowest point, which can be related to the vigorous economic development and encroachment of agricultural land and urban construction upon water bodies during this period. After 2011, the area of each water body type increased to some extent, which may be related to the policy asserting that "Lucid waters and lush mountains are invaluable assets" in this time period, when the government strengthened the protection of water bodies. The area of SWB changed frequently, a finding that is consistent with the typical climate change conditions in the region, and its area generally tended to increase, almost drawing level with the permanent water in 2020. The area of PWB, YLWB declined significantly during the study time, with $p < 0.01$. The $p$-values for the other two water types are greater than 0.05, which is not statistically significant.

The frequency area of [0–0.05] water bodies in Figure 8b is not available for 1988–1999 and 2012, but it is available in other years, which indicates that more details of the water bodies can be captured when the Landsat images are increased. The area of the [0, 1] water body range decreases, increases and then decreases in the 1984–2020 time period, with a peak in 2003, which may be related to the fact that Landsat7 started to malfunction in 2003 and the strips were lost. Although interpolation was performed to fill in the data gaps for this paper, much water body information was still lost.

### 3.4. Influence of Different Factors on Open-Surface Water Dynamics in the Study Area

### 3.4.1. Possible Natural Causes of Open-Surface Water Changes

By observing Figure S1 and Table S3, we can see that the interannual variation in the water body area and precipitation and temperature shows that the total annual precipitation and the average annual temperature and water area maintained a high degree of consistency. In Figure S1a, the total annual precipitation shows an overall increasing trend over the study period ($p = 0.0320$, Pearson's coefficient = 0.3690), with a significant positive correlation. In Figure S1b, the mean annual temperature shows a rapid increase ($p < 0.0001$, Pearson's coefficient = 0.7440), which indicates a highly significant positive correlation. This may be related to the background of "global warming". Table S3 shows the relationship of the area of each water body type with the total annual precipitation and the average annual temperature. All the water types are positively correlated with the total annual precipitation, especially SWB ($p = 0.035$, Pearson's coefficient = 0.3636) and SWB+YLWB ($p = 0.017$, Pearson's coefficient = 0.4075), which are significantly positively correlated, while the permanent water and year-long water are almost entirely uncorrelated with precipitation ($p > 0.05$). From the relationship between the temperature and various types of water bodies, we can observe that the area of each type of water was negatively correlated with temperature, and, surprisingly, there was a slight positive correlation between the area of SWB and temperature (Pearson's coefficient = 0. 1314).

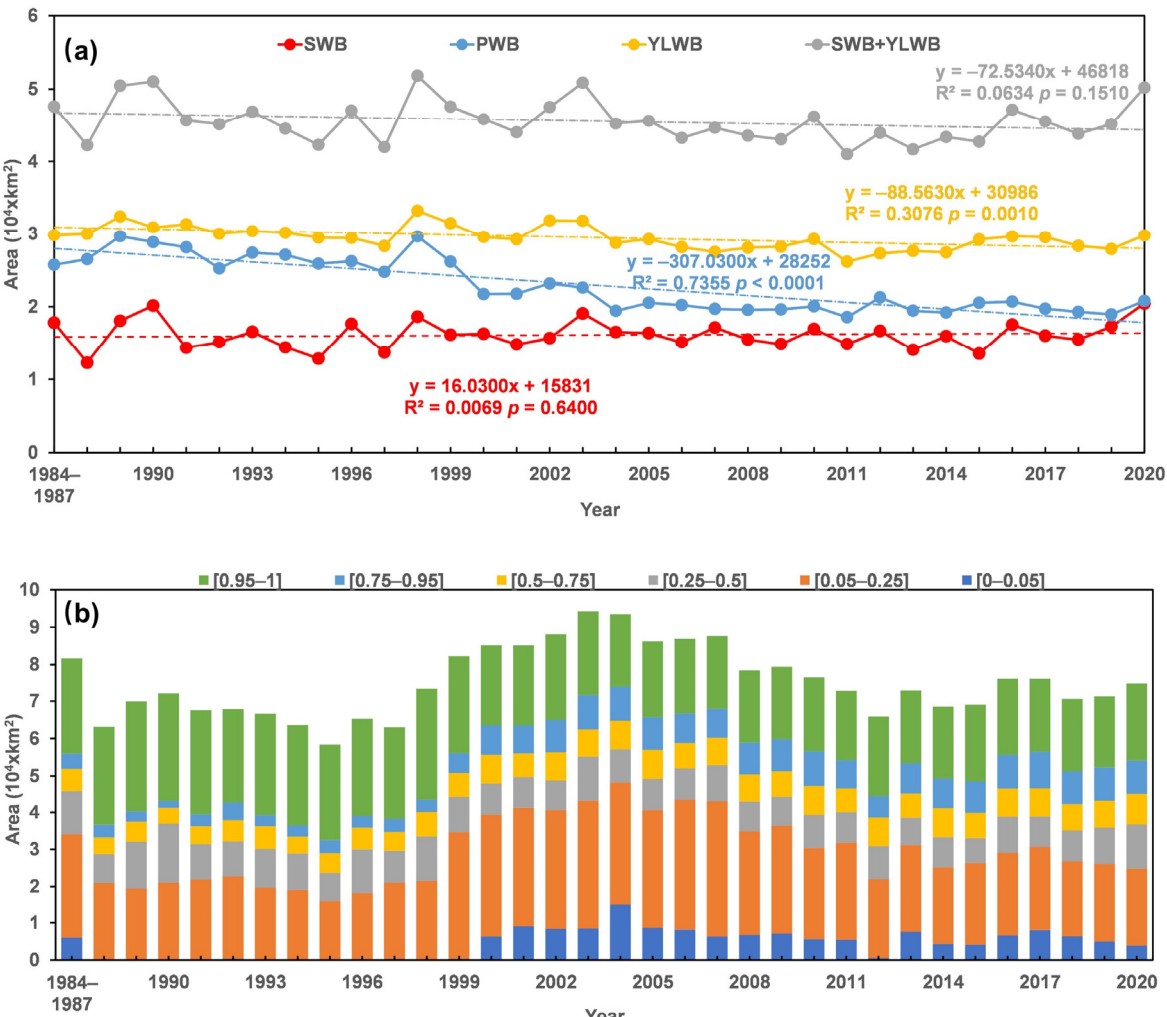

**Figure 8.** (**a**) The temporal variation in the surface water area for different water body types and (**b**) water area distribution for different water frequencies between 1984 and 2020 in the MLYP.

### 3.4.2. Possible Human Causes of Open-Surface Water Changes

From Figure S2, urban development can have a negative impact on the area of permanent water bodies. Seven provincial capitals, namely Wuhan, Changsha, Hefei, Nanchang, Nanjing, Hangzhou and Shanghai, were selected as the study area. Between 1984 and 2020, the expansion of the built-up areas of these cities was significant, and all these cities showed a significant decrease in the area of permanent water bodies ($p < 0.0001$). Shanghai, in particular, showed a significant linear decline, with $R^2 = 0.82$, mainly due to reclamation projects.

## 4. Discussion

### 4.1. Performance of the Water Extraction Methods

Nowadays, there are various water body index algorithms in use, but, in a study area with a complex environment, the water body spectra are very close to the vegetation and shadow spectra, resulting in water body pixels that cannot be completely distinguished from the vegetation and shadow pixels [47]. Thus, a single index cannot extract the water accurately in large-scale areas. In this study, we tested eight methods, including seven existing methods and the new method developed in this paper, in order to select the most accurate method for water body extraction, and the extraction results of four single indices were used to validate the abovementioned findings. Multi-index algorithms combining vegetation indices, such as the E-MVI method, can separate vegetation pixels, but the results

of this method show that the method resulted in great errors in extracting water from the Yangtze River basin, and the effect obtained by removing vegetation (rice) can similarly be achieved by using NDWI > −0.1 instead of EVI < 0.1. The newly developed N-MVI method in this paper with the NDWI > −0.1 condition can remove mountain shadows more effectively compared to the other two multi-index methods (MVI, A-MVI), and the method has the highest extraction accuracy for areas other than urban areas.

In the Yangtze River basin, one of the latest studies aimed to extract water using the MIWDR method, which is also known as the A-MVI method, as it is called in this paper. It was concluded that the MIWDR method can effectively mitigate the effects of urban buildings, although it can lead to the misclassification of some narrow urban rivers [17]. The results of this paper show that the method results in a loss of water body information not only for narrow urban rivers but also, in some cases, for large lakes.

Using the spectral index method alone inevitably leads to the suppression of some spectrally similar small water bodies when suppressing the effects of shading and vice versa. This phenomenon remains unresolved in the current research on improvements in water body index algorithms [22]. In this paper, we choose to incorporate products that can remove all kinds of shadows and combine them with the N-MVI method to obtain a balance between the suppression of shadows and maintenance of the extraction accuracy. Combining water body algorithms with impervious surface products and digital elevation model products helps to improve the accuracy of water body extraction in complex environments.

### 4.2. Attribution of Changes in Open-Surface Water Bodies in the MLYP from 1984 to 2000

Seasonal water has a significant positive relationship with the total annual precipitation, a weak relationship with PWB and YLWB, and a stronger relationship with SWB+YLWB. The results show that total annual precipitation affects other water types indirectly by directly increasing the amount of seasonal water, and precipitation is an important meteorological factor affecting the area of open-surface water bodies in the middle and lower reaches of Yangtze River. There is no doubt that another meteorological factor, average annual temperature, is negatively correlated with the water body area, but the results show that seasonal water is positively correlated with average annual temperature, partly because of climate anomalies in the study area and partly because this parameter may be related to human activities. The rice area is identified as SWB in this paper, and the area of rice cultivation in the study area is extensive [48]. Many studies have proven that human activities can have impacts on open-surface water bodies [33,36]. Through the results, it was found that the relationship between the PWB and temperature represents a more significant correlation compared to the relationships between other water types and temperature, possessing a strong correlation with urban expansion (Figure S2), and the impact of human activities on permanent water can be clearly observed through the changes in the Wuhan lakes complex from 1990 to 2020 (Figure S3). Human activities do not always have a negative impact on water bodies. For example, the construction of dams can change the distribution of water bodies. After the completion of the Three Gorges Dam in 2003, the water area west of the dam gradually increased (Figure S4). However, Three Gorges Dam impoundment can also negatively affect various water bodies in the MLYP east of the dam, lowering lake levels and causing severe droughts.

### 4.3. Comparison with Similar Studies

#### 4.3.1. Comparison with the JRC Global Surface Water Mapping Layers

In this section, our WF results are compared with the 'occurrence' layer of JRC [2]. JRC Global Surface Water Mapping Layers v1.3 is a water body dataset with a 30 m spatial resolution generated through Landsat 5/7/8 images taken from 1984 to 2020. In general, the spatial distribution patterns of open-surface water bodies are consistent between the two datasets, especially in the [25–75], [75–100] frequency range (Figure 9). However, due

to the differences between the water extraction methods, the area differences between the two datasets in the [0–5] interval are quite large (Figure S5).

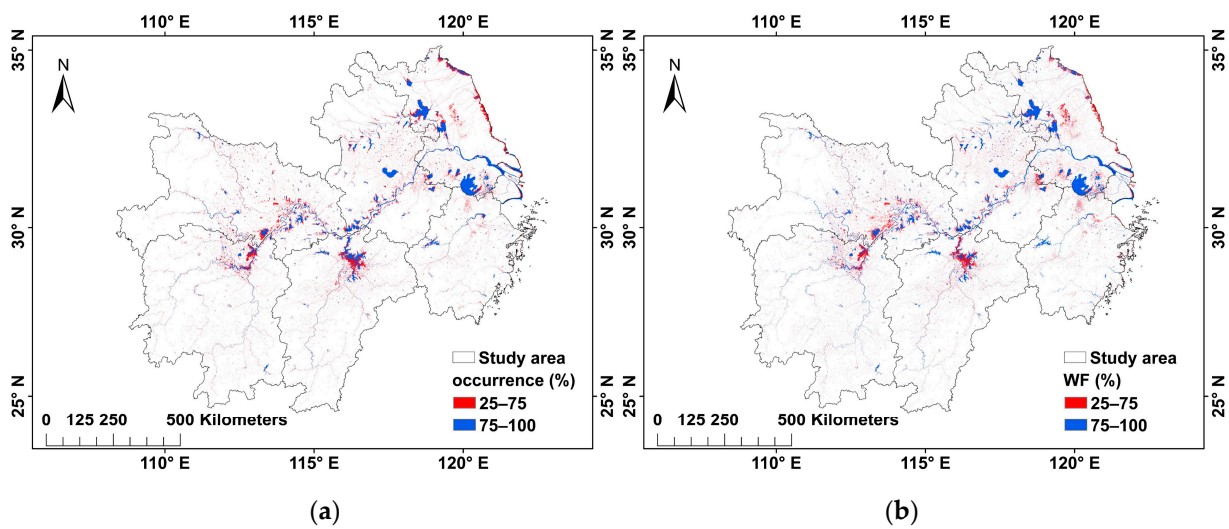

(**a**)                                      (**b**)

**Figure 9.** (**a**) JRC Global Surface Water Mapping Layers; (**b**) WF dataset of this paper for the MLYP from 1984 to 2020.

### 4.3.2. Comparison with JRC Yearly Water Classification History v1.3

In this subsection, we compare the annual water classification dataset obtained in this study with the JRC yearly water classification historical dataset v1.3. In general, the temporal trends of open-surface water bodies are consistent between the two datasets, and there are some differences in the areas of seasonal water and permanent water between the two datasets due to their different classification criteria. Such results are normal. However, in some years (the years with red dots in Figure 10), the results of the two datasets differed greatly, and, by contacting the JRC in 1998 (Figure 11a), we found that JRC could not completely cover the study area in these years, which led to incorrect results. The annual water body products obtained using the methods explored in this paper fill the gaps in the water body area data of the MLYP for these years.

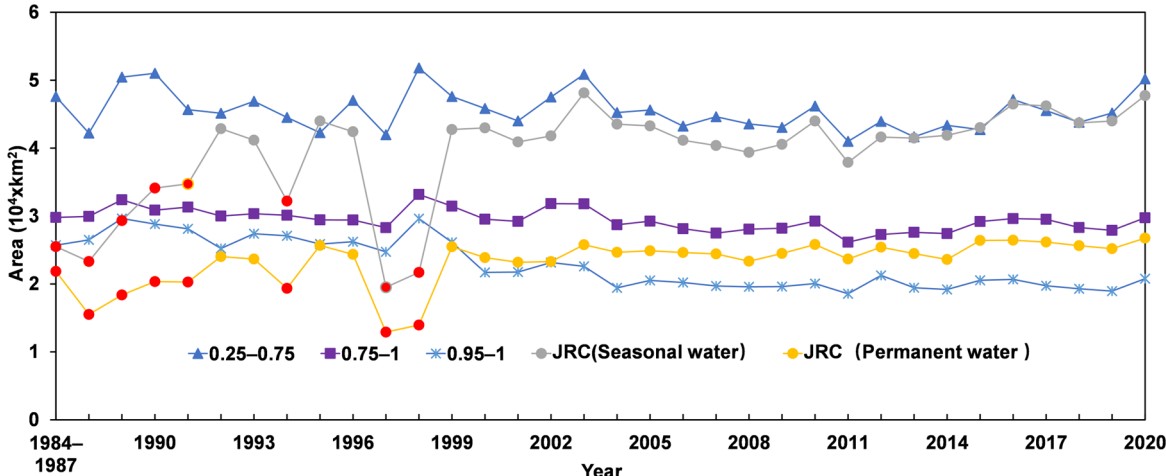

**Figure 10.** Comparison of the MLYP's yearly open water body area with the JRC v1.3 data (the red dots in the figure represent the yearly data of the JRC errors in the MLYP).

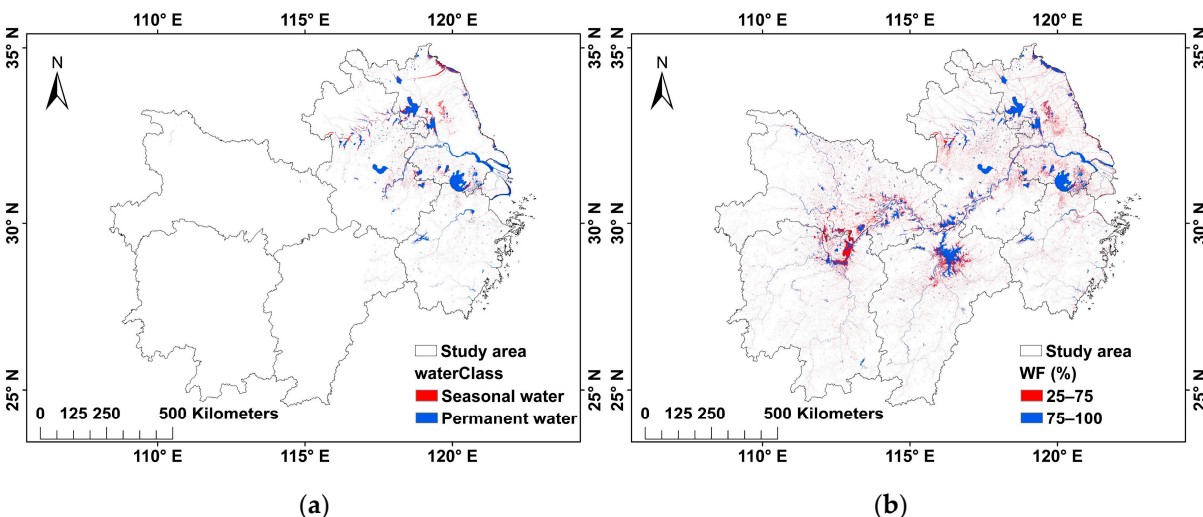

**Figure 11.** (**a**) JRC yearly water classification historical data; (**b**) WF year data of this paper for the MLYP in 1998.

### 4.4. Limitations and Future Improvements

In this study, we used Landsat data to produce a dataset of water bodies in MLYP with a 30 m spatial resolution and 36-year temporal resolution, which fills the gap in the JRC data for the water bodies in the region in the 1980s and 1990s. Through the water frequency, we were able to analyze the inter- and intra-annual trends of the water bodies, but the MLYP has high rainfall in the summer and the water bodies change seasonally and can even change rapidly between different months. Moreover, in the summer, the clouds are thick, and, as Landsat images are easily affected by clouds, this limits the real-time high-frequency monitoring of open-surface water bodies when extreme precipitation occurs, resulting in inaccurate results. This problem is solved by adding other data for the study of multi-source remote sensing spatial and temporal fusion, such as MODIS, with a daily temporal resolution, Sentinel-1 synthetic aperture radar (SAR), Sentinel-2, GF, Planet and other high-resolution data, especially SAR images, which can penetrate clouds and can be used in cloudy weather.

The method of extracting water bodies explored in this paper can be used for three typical areas in the MLYP, but, as mentioned above, the water bodies in this area have large intra-annual variation and using an annual-scale impervious surface to remove the building noise will cause the water body results to be inaccurate. However, the water body index method alone cannot extract water bodies on complex features. Yang et al. used GHSL [14] data to calculate water bodies individually for natural areas and built-up areas, and, subsequently, we can follow this method to select data to calculate water bodies separately for built-up areas and natural areas over the past 36 years.

In the section on the impact of human activities on water bodies, we discussed the impacts of cities and dams on water body changes, but there are many other factors that affect water bodies in this highly populated region that have not been analyzed, such as the expansion of paddy fields due to agricultural production. The region has abundant rainfall and is suitable for growing rice, and it comprises five of the top six provinces in China in terms of rice production. Rice fields are easily identified as seasonal water and confused with water bodies. In order to distinguish the two, we can use the phenological method to extract rice in time periods based on the product of multi-source remote sensing data fusion. There are many scholars who have conducted such studies in locations such as the Northeast Plain [49] and Japan [50]. However, the method used in these studies cannot be used directly for the middle and lower reaches of the Yangtze River plain due to the difference in climate zone. Subsequently, we used this concept to carry out research in order to calculate the extent of paddy fields, with the aim of carrying out analytical research on the impact of paddy fields on water bodies.

## 5. Conclusions

The main conclusions of this paper are twofold. Firstly, a new water body extraction method was proposed. The current commonly used multi-indicator water body index method was improved and combined with impervious surface data and DEM data to maximize the retention of water body information and reduce the misclassification of highlighted and dark building surfaces, building shadows, mountain shadows, vegetation, etc. The method is highly accurate, easy to implement and suitable for open-surface water body extraction in large-scale, complex environments.

Secondly, based on this method, we constructed a 30 m scale open-surface water body dataset for the middle and lower reaches of the Yangtze River from 1984 to 2020 on the GEE platform based on high-quality Landsat 5/7/8 images and analyzed the inter- and intra-annual dynamic characteristics of open-surface water bodies in the study area. We classified the open-surface water bodies into TWB, TWBS, SWB, YLWB and PWB according to the frequency of the water bodies. PWB and YLWB showed a significant decreasing trend over time. Finally, we explored the relationship between each driving mechanism and the water area. Regarding the climate factors, precipitation was positively correlated with the water area, especially in the case of SWB and SWB+YLWB, which were significantly positively correlated with precipitation. With the exception of SWB, temperature was negatively correlated with the area of the other types of water bodies, and the anomalous relationship between SWB and temperature may be related to seasonal rice. PWB was highly significantly correlated with temperature because, in addition to temperature, urban development has had a significant effect on the PWB area.

**Supplementary Materials:** The following supporting information can be downloaded at: https://www.mdpi.com/article/10.3390/rs15071816/s1, Table S1: The Landsat images for accuracy assessment. Table S2: Accuracy of water body extraction under different water body indices in different study backgrounds. Figure S1: Annual variations in (a) the water area and precipitation and (b) the water area and temperature based on the area of each open-surface water type from 1984 to 2020 in the MLYP. Table S3: Pearson correlation coefficients and *p*-values between the surface water area and precipitation and temperature in the MYLP between 1984 and 2020. Figure S2: The temporal variation and linear trend regression of open-surface permanent water area for seven provincial capitals in the MLYP. Figure S3: The decreases in water bodies due to urban development. Figure S4: The increases in water bodies due to the construction of the Three Gorges Dam hydraulic facility. Figure S5: Comparing the area of each classified water frequency in JRC and this paper.

**Author Contributions:** Conceptualization, W.W. and H.T.; methodology, W.W. and H.T.; software, W.W. and L.Z.; validation, W.W.; data curation, W.W. and L.H.; writing—original draft preparation, W.W.; writing—review and editing, H.T. All authors have read and agreed to the published version of the manuscript.

**Funding:** This research was supported by the National Natural Science Foundation of China (No. 41901055). We thank all the people involved in modelling and data analysis related to this study.

**Data Availability Statement:** Not applicable.

**Acknowledgments:** We acknowledge Google Earth Engine for providing computing platform. The free Landsat, Sentinel, GAIA, JRC-GSW, HAND, GMTED2010 used in this study are all available in GEE. We also thank those anonymous reviewers who provide valuable suggestions for the revision of this paper.

**Conflicts of Interest:** The authors declare no conflict of interest.

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
