# Peer review of "Long-Term Changes in Water Body Area Dynamic and Driving Factors in the Middle-Lower Yangtze Plain Based on Multi-Source Remote Sensing Data"

_remotesensing, doi:10.3390/rs15071816_

Round 1
Reviewer 1 Report
This paper mainly focusses on water bodies extraction using the water index method based on Landsat satellite images in a long time series over a wide-ranging area.
The quality of the research is qualified and significant for understanding. The paper is recommended for publication after fixing some major improvements of major revision. Notably, the paper needs to explain the problem how water extraction methods works and how to evaluate the method. Comments follow.
#1. Line242-245, please clarify how this method improved and why it works.
#2. Eq(4), how 2.5 and 7.5 be determined?
#3. Line467-468, “during this time period, with p < 0.0001, and the year-long water also declined, with p = 0.001.” Here needs to explain how these two points are choosed?
#4. The author should take the influence of Three Gorges Dam impoundment, which has significant impact on the TWS in the MLYP.
#5. The full text should be carefully refined and smoothed.
#6. Some figures in the manuscript needs to improve the quality according to the journal’s guide
Author Response
Dear reviewer:
Thank you very much for your comments and professional advice. These opinions help to improve academic rigor of our article. Based on your suggestion and request, we have made corrected modifications on the revised manuscript. We hope that our work can be improved again. Furthermore, we would like to show the details as follows:
Point 1: #1. Line242-245, please clarify how this method improved and why it works.
Response 1: We apologize for not clarifying this method, the method in this paper was obtained by our continuous testing of each index in the GEE platform, but this was not convincing. We redid the test experiment and made box plots by selecting five types of sample points, and proved that the interference of mountain shadows and vegetation could be minimized under (NDWI > -0.1) and (MNDWI > NDVI or MNDWI > EVI) conditions. For details, please refer to lines 356-414 of the revised manuscript.
Point 2: #2. Eq(4), how 2.5 and 7.5 be determined?
Response 2: Please forgive us for not adding the EVI reference in the introduction section, the formula for this vegetation index we cited from this article [1].
Point 3: #3. Line467-468, “during this time period, with p < 0.0001, and the year-long water also declined, with p = 0.001.” Here needs to explain how these two points are choosed?
Response 3: These two points were chosen because there are statistically significant differences between these two types of water bodies. There is a statistically significant difference when p-value < 0.05, and an extremely significant statistical difference when p < 0.01.
Among these four water types, PWB has a p-value < 0.0001, indicating a highly significant statistical difference between permanent water and time, and k is negative, so the area of PWB is highly significantly decreased during the study time. YLWB has a p-value = 0.001 < 0.01, and k is negative, so the area of YLWB is significantly decreased during the study time. The p-values for the other two water types are greater than 0.05, which is not statistically significant, so these two other points were not selected in this paragraph.
Point 4: #4. The author should take the influence of Three Gorges Dam impoundment, which has significant impact on the TWS in the MLYP.
Response 4: We thank the reviewer for pointing this out. does the TWS in this sentence refer to transient water bodies that tend to be seasonal? The Three Gorges Dam does have an impact on TWBS in the middle and lower reaches of the Yangtze River. The Three Gorges Dam was chosen in our article to explain that human activities would change the distribution of water bodies. We used pictures to show the change in the area of the water bodies in the western part of the dam before and after the dam was built, and did not analyze the effect of water impoundment on the water bodies in the eastern part of the dam, and we added a sentence at the end. “However, Three Gorges Dam impoundment can also negatively affect various water bodies in the MLYP east of the dam, lowering lake levels and causing severe droughts.”
Point 5: #5. The full text should be carefully refined and smoothed.
Response 5: We have refined and smoothed out the entire text within the best of our ability, but due to limited capacity, the result may not meet the requirements of the revision, for which we deeply apologize.
Point 6: #6. Some figures in the manuscript needs to improve the quality according to the journal’s guide
Response 6: Thank you for the reminder. By observation we found that some elements in Figure 7 and Figure 10 were not very clear and we reworked them. We found no obvious problems with the other figures, and all figures are 300 dpi resolution. If there are still figures in the text that do not meet the requirements, please help us point them out. We would be grateful for this.
Thank you very much for your attention and time. Look forward to hearing from you!
References
- Huete, A.D., K.; Miura, T.; Rodriguez, E.P.; Gao, X.; Ferreira, L.G Overview of the radiometric and biophysical performance of the MODIS vegetation indices Remote Sens. Environ 2002, 83, 195-213, doi:10.1016/S0034-4257(02)00096-2.

Reviewer 2 Report
The article attempts to present an indices method of detecting water bodies using Landsat imagery and then delineate long-term trends in water bodies in the study area. The article needs revision to make it suitable for publication.
I got confused reading the article when it comes to the key message in the paper. Did the authors attempt to develop a new method of detecting pixels representing water bodies? Did the authors attempt to compare various indices for detecting pixels representing water bodies? Did the authors attempt to carry out a trend analysis of parameters in water bodies? If so, which specific parameters were they interested in (surface area, volume, morphology, distribution……)? Were they attempting to do all of these? If so, the paper would need a major revision to clearly layout the story so that the reader can follow the flow.
Please see detailed review attached.

Author Response
Dear reviewer:
Thank you very much for your comments and professional advice. These opinions help to improve academic rigor of our article. Based on your suggestion and request, we have made corrected modifications on the revised manuscript. We hope that our work can be improved again. Furthermore, we have provided a point-by-point response to your comments and uploaded it as a Word file.

Reviewer 3 Report
The paper is interesting from the point of view of detecting water bodies from Landsat images. I would suggest adding more details on meteorological variables. It is mentioned precipitation and temperature but we do not know about the time step of these and the origin of data (from the met stations or averaged spatially?) How important is the seasonal variability of the water bodies areas ? The study area covers a very big region with a variable climate and monsoon type of precipitation patterns.

Author Response
Dear reviewer:
Thank you very much for your comments and professional advice. These opinions help to improve academic rigor of our article. Based on your suggestion and request, we have made corrected modifications on the revised manuscript. We hope that our work can be improved again. Furthermore, we would like to show the details as follows:
Point 1: The paper is interesting from the point of view of detecting water bodies from Landsat images. I would suggest adding more details on meteorological variables. It is mentioned precipitation and temperature but we do not know about the time step of these and the origin of data (from the met stations or averaged spatially?) How important is the seasonal variability of the water bodies areas ? The study area covers a very big region with a variable climate and monsoon type of precipitation patterns.
Response 1: Our details on precipitation and temperature meteorological variables are in the fifth of 2.2.2 Supplementary Data. The two data on precipitation and temperature are obtained from the 'Chinese Meteorological Elements Day-by-Day Station Observation Dataset' of the Resource and Environment Science and Data Center of the Chinese Academy of Sciences (https: www.resdc.cn), a dataset obtained from daily value observations of more than 2400 meteorological stations in China. There are 507 meteorological stations in our study area, and the daily mean temperature and daily 20-20 h cumulative precipitation from these meteorological stations were selected for the period 1984-2020. In this paper, we attempt to analyze the water body area changes on an annual scale, and the climate data should also be treated as an annual scale. We summarize the daily average temperature for the whole year and then divide it by one year to get the annual average temperature, and summarize the daily 20-20 hour cumulative precipitation for the whole year to get the annual total precipitation.
Point 2: negative units of DEM ?
Response 2: The elevation values in the DEM are relative to the reference altitude, so a negative value does not necessarily mean that the ground height is lower than sea level, but rather that the altitude of the area is lower than the reference altitude.
Point 3:10m
Response 3: We apologize for the error, the spatial resolution of Sentinel 2 is 10m, which has been corrected in the article.
Point 4: Figure include Rule 1-4 which are explained later. Please relocate the figure below the Table 1.
Response 4: We thank the reviewer for pointing this out. We have placed Figure 3 below Table 1.
Point 5: annual totals ?
Response 5: We have corrected this error by adding ‘total annual’ in front of the precipitation.
Thank you very much for your attention and time. Look forward to hearing from you!
Round 2
Reviewer 1 Report
I think all the points are clearly modified and the manuscript should be accepted.
Reviewer 2 Report
The authors agreed to all of my suggestions. Therefore, I dd not have any further objections. The Editor may allow publications after ensuring that all suggestions are addressed, and all minor grammatical errors are corrected.
Reviewer 3 Report
OK text is properly improved.